Subject Area:
cellular biology

Keywords:
EZN-2208, chronic lymphocytic leukaemia, fludarabine, hypoxia-inducible factors

Author for correspondence:
Rosa Bernardi
e-mail: bernardi.rosa@hsr.it

†Present address: San Raffaele Telethon Institute for Gene Therapy (SR-TIGET), IRCCS San Raffaele Scientific Institute, Milan, Italy.

# EZN-2208 treatment suppresses chronic lymphocytic leukaemia by interfering with environmental protection and increases response to fludarabine

Roberta Valsecchi[1], Nadia Coltella[1,†], Daniela Magliulo[1], Lucia Bongiovanni[2], Lydia Scarfò[1,3], Paolo Ghia[1,3], Maurilio Ponzoni[2,3] and Rosa Bernardi[1]

[1]Division of Experimental Oncology, and [2]Pathology Unit, IRCCS San Raffaele Scientific Institute, Milan, Italy
[3]Vita-Salute San Raffaele University School of Medicine, Milan, Italy

RB, 0000-0002-3607-6336

The transcription factor HIF-1α is overexpressed in chronic lymphocytic leukaemia (CLL), where it promotes leukaemia progression by favouring the interaction of leukaemic cells with protective tissue microenvironments. Here, we tested the hypothesis that a pharmacological compound previously shown to inhibit HIF-1α may act as a chemosensitizer by interrupting protective microenvironmental interactions and exposing CLL cells to fludarabine-induced cytotoxicity. We found that the camptothecin-11 analogue EZN-2208 sensitizes CLL cells to fludarabine-induced apoptosis in cytoprotective *in vitro* cultures; *in vivo* EZN-2208 improves fludarabine responses, especially in early phases of leukaemia expansion, and exerts significant anti-leukaemia activity, thus suggesting that this or similar compounds may be considered as effective CLL therapeutic approaches.

## 1. Introduction

Chronic lymphocytic leukaemia (CLL) is a monoclonal disorder of mature B cells that accumulate in the peripheral blood (PB), bone marrow (BM) and lymphoid organs [1]. CLL has a highly variable disease course, from an indolent disease that requires no immediate intervention to a more aggressive disorder that necessitates early treatment [2]. Treatment options vary based on genetic features of the leukaemic cells, especially aberrations of the TP53 gene that predict poor response to therapy. Based on improved understanding of CLL biology, standard therapy with fludarabine, cyclophosphamide and the anti-CD20 antibody rituximab has recently been supplemented with new agents, like inhibitors of B-cell receptor signalling (ibrutinib and idelalisib) and BCL-2 (venetoclax), which demonstrate impressive efficacy also in high-risk patients [3]. Nevertheless, the quest for a cure and for eradication of minimal residual disease (MRD) is still enduring as most patients eventually relapse and need additional lines of therapy.

We have recently demonstrated that the transcription factor HIF-1α promotes CLL maintenance by supporting heterotypic interactions with stromal cells in protective microenvironments that are sites of MRD persistence [4]. In this study, we aimed to test if HIF-1α inhibitors may act as chemosensitizers by promoting loss of contact between CLL cells and protective BM stroma. With this aim, we used the PEGylated camptothecin-11 analogue EZN-2208 to inhibit HIF-1α [5]. Like most inhibitors of transcription factors, EZN-2208 is not a specific HIF-1α inhibitor, and displays cytotoxic activity mainly due to the inhibition of topoisomerase I. However, we and others have previously shown that EZN-2208 efficiently inhibits HIF-1α even when used at concentrations that do not trigger cell death in a topoisomerase I-dependent manner [4,5]. In leukaemia

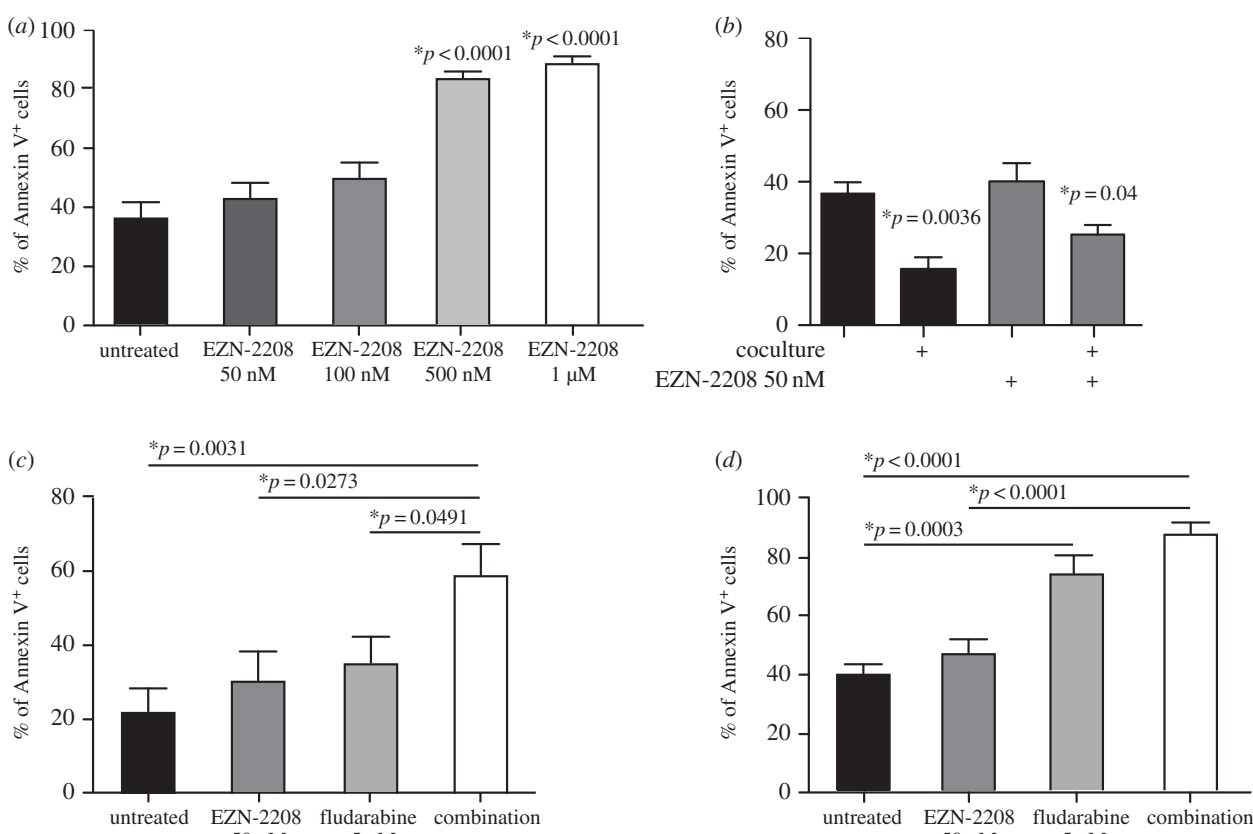

**Figure 1.** EZN-2208 neutralizes the protective effect of BM stroma and improves fludarabine cytotoxic activity on CLL cells. (a) Percentage of Annexin V+ primary CLL B cells treated with the indicated doses of EZN-2208 for 48 h. Data represent mean values ± s.e.m. of 11 independently processed primary samples. p-values are calculated versus untreated cells. (b) Percentage of Annexin V+ primary CLL B cells in the presence or absence of HS5 stromal cells upon 48 h treatment with EZN-2208 at the indicated concentration. Data represent mean values ± s.e.m. of eight independently processed primary samples. p-values are calculated versus untreated cells. (c) Percentage of Annexin V+ primary CLL B cells in co-culture with HS5 cells upon 48 h treatment with the indicated agents. Data represent mean values ± s.e.m. of eight independently processed primary samples. (d) Percentage of Annexin V+ primary CLL B cells treated with the indicated agents for 48 h. Data represent mean values ± s.e.m. of eight independently processed primary samples.

settings, by adjusting EZN-2208 dosage to non-cytotoxic concentrations, we have demonstrated that this compound can impair myeloid leukaemia and CLL progression *in vivo* via HIF-1α inhibition [4,6]. In the current work, to unveil increased sensitivity to the cytotoxic agent fludarabine, we selected a concentration of EZN-2208 (50 nM) that did not induce cell death in primary CLL cells, both when cultured alone and cocultured with BM-derived stroma (figure 1*a,b*), to mimic the protective BM microenvironment, a condition that knowingly reduces intrinsic and drug-induced apoptosis [7]. Notably, 50 nM EZN-2208 did not induce apoptosis in stromal cells (not shown). Interestingly, EZN-2208 significantly increased fludarabine-induced mortality only in the protective co-culture condition (figure 1*c*), whereas in the absence of stroma adding EZN-2208 did not significantly improve fludarabine response (figure 1*d*). These results indicate that EZN-2208 sensitized CLL cells to apoptosis by interfering with protective microenvironmental interactions and may add therapeutic efficacy to current CLL treatments by impairing such protective cell contacts.

To evaluate if EZN-2208 acted as a chemosensitizer *in vivo*, we exploited a transplantable model of *Eµ-TCL1*-derived CLL [4]. To mimic clinical protocols, we choose a slow-progressing *Eµ-TCL1*-derived CLL, and subjected mice to consecutive treatment cycles. Thirty-four days after inoculation of leukaemic cells, when leukaemia burden reached 50% in PB, mice were administered a first cycle of treatment with EZN-2208

(5 mg kg⁻¹), fludarabine (34 mg kg⁻¹) or a combination of the two drugs, both used at concentrations previously characterized in *Eµ-TCL1* mice [4,8]. One cohort of mice was sacrificed at the end of treatment (day 43), and significant reduction in leukaemia burden in BM and spleen was observed only upon combined EZN-2208 and fludarabine treatment, while the single compounds exerted no or minor effects (figure 2*a*). A second cohort of mice was subjected to an additional treatment cycle at day 54 after leukaemia inoculation, when leukaemic cells in PB reached 50–70%, and mice were sacrificed at the end of treatment (day 63). Interestingly, after the second cycle of treatment, EZN-2208 efficiently reduced disease burden while fludarabine showed no effect, and combination treatment mimicked the effect of EZN-2208 (figure 2*b*). Consequently, co-treatment with EZN-2208 and fludarabine did not prolong mice survival when compared with EZN-2208 treatment alone (not shown). These results show that in a slow-progressing CLL model EZN-2208 and fludarabine act cooperatively especially at early phases of disease expansion, and further confirm that EZN-2208 treatment efficiently reduces leukaemia progression, in line with a previous report of ours, where we had observed that EZN-2208 treatment reduced leukaemia burden after a single treatment cycle [4].

To confirm these results in a mouse model of aggressive CLL with dysfunctional p53 [9], we transplanted the human CLL cell line MEC-1 in *Rag2⁻/⁻γc⁻/⁻* mice, a model that is

**Figure 2.** *In vivo* EZN-2208 treatment exerts significant anti-leukaemia effects and partly improves response to fludarabine. (*a*) (i) Spleen weight of C57B6 mice injected with EµTCL1-derived leukaemia, treated with one cycle of the indicated agents and sacrificed at the end of treatment. Data represent mean values ± s.e.m. (*n* = 3). (ii) Percentage of leukaemic cells (calculated as CD5$^+$CD19$^+$ cells over total cells) in the BM of transplanted mice treated with indicated agents as in (*a*). Data represent mean values ± s.e.m. (*n* = 3). (iii) Percentage of leukaemic cells (calculated as CD5$^+$CD19$^+$ cells over total cells) in the PB of mice treated with indicated agents as in (*a*). Data represent mean values ± s.e.m. (*n* = 3). (*b*) (i) Spleen weight of transplanted leukaemic mice treated with two cycles of the indicated agents and sacrificed at the end of the second cycle of treatment. Data represent mean values ± s.e.m. (*n* = 3). (ii) Percentage of leukaemic cells (calculated as CD5$^+$CD19$^+$ cells over total cells) in the BM of transplanted mice. Data represent mean values ± s.e.m. (*n* = 3). (iii) Percentage of leukaemic cells (calculated as CD5$^+$CD19$^+$ cells over total cells) in the PB of transplanted mice. Data represent mean values ± s.e.m. (*n* = 3). (*c*) Percentage of MEC-1 leukemic cells (calculated as CD19$^+$ cells over total cells) in the spleen (i), BM (ii) and PB (iii) of mice treated with the indicated agents at day 18 post-transplantation and sacrificed at the end of treatment. Data represent mean values ± s.e.m. (*n* = 3). (*d*) Kaplan–Meier survival curve of *Rag2$^{-/-}$γc$^{-/-}$* mice injected with MEC-1, treated as indicated and sacrificed when terminally sick (*n* = 6). Significant *p*-values calculated by log-rank (Mantel–Cox) test are as follows: EZN-2208 versus untreated, *p* = 0.0286; EZN-2208 versus fludarabine, *p* = 0.0075; combination treatment versus untreated, *p* = 0.0045; combination treatment versus fludarabine, *p* = 0.0005.

**Table 1.** Clinical characteristics of primary CLL samples. M, male; F, female; n.a., not available; del, deletion. For IGHV identity: M, mutated (less than or equal to 98%); U, unmutated (greater than 98%).

| CLL# | age at diagnosis | gender | CD38 (PB%) | Rai stage at diagnosis[a] | IGHV identity | disease course | cytogenetic |
|---|---|---|---|---|---|---|---|
| 138 | 49 | F | 3 | low | M | progressive | n.a. |
| 050 | 60 | M | 11 | low | U | progressive | del(11q), del(13q) |
| 370 | 57 | M | 2.3 | low | U | stable | del(13q) |
| 236 | 70 | M | 0.04 | low | M | stable | del(13q) |
| 250 | 54 | M | 19.1 | low | n.a. | stable | n.a. |
| 219 | 58 | M | 1.2 | low | U | progressive | del(17q), del(13q) |
| 242 | 64 | F | 0.1 | low | n.a. | stable | del(13q) |
| 123 | 62 | F | 0.2 | low | M | stable | del(13q) |
| 225 | 74 | F | 15.9 | low | M | progressive | normal |
| 489 | 50 | M | 7 | Int | M | progressive | n.a. |
| 117 | 45 | M | 0.3 | low | U | progressive | del(13q) |
| 136 | 66 | M | 33.8 | low | M | stable | del(17q), del(13q) |
| 019 | 70 | F | 0 | low | M | stable | del(13q) |
| 186 | 61 | M | 0 | low | M | stable | normal |
| 247 | 74 | F | 0.3 | low | M | stable | n.a. |
| 316 | 64 | M | 24.7 | Int | M | progressive | trisomy 12 |
| 509 | 52 | F | 0.4 | low | M | stable | n.a. |
| 197 | 38 | M | 0.2 | low | M | stable | normal |

[a]Low, Stage 0; Intermediate, Int, Stages I and II; High, Stages III and IV.

insensitive to fludarabine treatment. To assess whether EZN-2208 treatment sensitized MEC-1-driven CLL to fludarabine, we first used the dose of 5 mg kg$^{-1}$ EZN-2208, which we had previously characterized in this mouse model [4]. Similar to previous experiments, EZN-2208 treatment effectively slowed CLL progression; however, we did not observe any additive effects upon adding fludarabine (data not shown). We thus hypothesized that diminishing EZN-2208 efficacy may unveil possible additive effects of fludarabine, and treated MEC-1-transplanted mice with 2 mg kg$^{-1}$ EZN-2208. Even if used at a lower concentration, EZN-2208 significantly impacted leukaemia progression in this aggressive CLL model, while fludarabine treatment had no effect (figure 2c,d). Adding fludarabine to EZN-2208 further reduced CLL involvement in the BM (figure 2c), although combination treatment did not improve mice survival with respect to EZN-2208 treatment alone (figure 2d).

In summary, our studies confirm that EZN-2208 is an effective compound that suppresses CLL progression in different CLL mouse models, including a disease with adverse prognostic markers that appears refractory to treatment with fludarabine. This conclusion is in accordance with a recently published work where another compound targeting HIF-1α was found to exert anti-tumour activities in p53-mutated CLL [10].

Surprisingly, we observed that although EZN-2208 added therapeutic value to fludarabine treatment *in vitro*, in co-cultures where CLL cells are protected from cell death by BM-derived stromal cells (figure 1), in the more complex *in vivo* microenvironment EZN-2208 sensitized CLL cells to fludarabine only partially. One possible explanation is that fludarabine is poorly effective *in vivo* at the concentration used in our experiments, or in the mouse models that we used, although we selected a concentration previously characterized in *Eμ-TCL1* mice [8]. Nonetheless, the main conclusion of our work is that EZN-2208 exerts strong anti-CLL activities in two *in vivo* systems. The effectiveness of EZN-2208 may be due to a number of *in vivo* functions, besides its cytotoxic activity. For instance, we previously reported that EZN-2208 inhibits neo-angiogenesis in CLL mouse models [4]. In addition, because HIF-1α is an important regulator of immune cell functions [11], EZN-2208 may also interfere with the supporting action of lymphoid or myeloid immune regulators that promote CLL maintenance and proliferation [12]. Interestingly, our experiments show that EZN-2208 targets especially CLL populations residing in BM and spleen (figure 2). Because CLL cells express higher levels of HIF-1α when in contact with stromal cells [4,13], our data suggest that CLL cells residing in protective niches rely on HIF-1α-dependent pro-survival signals more significantly than cells in peripheral circulation.

In conclusion, our work suggests that pharmacological strategies aimed at inhibiting HIF-1α may be of added value for CLL therapy, and further studies should be performed to evaluate the efficacy of these compounds in settings that recapitulate drug-resistant disease for future clinical development.

## 2. Methods

MEC-1 and HS5 cells (DSMZ and ATCC) were maintained in RPMI-1640 and DMEM supplemented with 10% FBS and 1% Pen-Strep antibiotics (Lonza) at 37°C in a humidified atmosphere containing 5 and 10% $CO_2$. CLL patients (clinical

features shown in table 1) were diagnosed per International Workshop on CLL (iwCLL) guidelines [1], and were either untreated or off therapy for at least six months. Leukaemic CD19$^+$ cells obtained with informed consent as approved by the institutional ethics committee at San Raffaele Hospital were used immediately after isolation with RosetteSep Human B Cell Enrichment Cocktail and Lymphoprep (STEM-CELL Technologies). EZN-2208 was provided by Belrose Pharma Inc., fludarabine purchased from Sandoz and CMFDA from Life Technology.

For co-culture experiments, $3 \times 10^6$ PB-derived CLL cells were labelled with 1 μM CMFDA, added to a HS5 monolayer and treated with EZN-2208 and fludarabine. Forty-eight hours later, non-adherent CLL cells were collected and cell viability was evaluated by flow cytometry as percentage of Annexin V$^+$ cells over the total number of CMFDA$^+$ cells.

$Rag2^{-/-}\gamma c^{-/-}$ and C57BL/6 $E\mu$-TCL1 mice were maintained in specific pathogen-free animal facilities and treated in accordance with European Union and Institutional Animal Care and Use Committee (IACUC) guidelines. C57BL/6 mice were injected i.p. with $10 \times 10^6$ splenic cells from $E\mu$-TCL1 leukaemic mice; $Rag2^{-/-}\gamma c^{-/-}$ mice were injected i.v. with $10 \times 10^6$ MEC-1 cells. When indicated, mice were treated intravenously (i.v.) with five administrations of EZN-2208 every other day, and/or intraperitoneally (i.p.) with fludarabine for 5 consecutive days starting on day 3 of EZN-2208 administration.

Immunophenotypic analysis was performed with the following antibodies: anti-human CD19 (PC-7) from Beckman Coulter, anti-mouse CD5 (APC) and anti-mouse CD19 (PECY-7) from BD Biosciences. Annexin V staining was performed using the PE Annexin V Apoptosis Detection Kit I (BD Pharmigen).

For survival experiments, the Kaplan–Meier curves were analysed with the Mantel–Cox test. Unless otherwise stated, two-sided Student's $t$-test was used to measure statistical significance.

Ethics. The experiments presented in this publication have been approved by institutional and governmental committees. Animal studies were approved by the San Raffaele Hospital Institutional Animal Care and Use Committee (IACUC) and by the central committee that revises animal experimentation at the Italian Ministry of Health. Collection of cells from patients at San Raffaele Hospital was obtained upon collection of informed consent forms and was approved by the Institutional Ethics Committee of San Raffaele Hospital in accordance with National and International guidelines.

Data accessibility. This article has no additional data.

Authors' contributions. N.C. evaluated HIF-1α expression in patients' cells; D.M. measured CLL cells in leukaemic mice; L.B. and M.P. performed histopathological evaluations of leukaemic mice; L.S. and P.G. designed experiments with patients' cells; all other experiments were performed by R.V. R.B. and R.V. analysed the data and wrote the paper. All authors discussed the results and commented on the manuscript.

Competing interests. We declare we have no competing interests.

Funding. This work was supported by Italian Association for Cancer Research (AIRC, Special Program Molecular Clinical Oncology, 5 per mille #9965 to P.G.), a CLL Global Research Foundation Grant to R.B. and a Lady Tata Memorial Trust Award to R.V.

Acknowledgements. The authors would like to thank all current and previous members of the laboratory for valuable discussion and support, and L. Greenberger, Y. Zhang and Belrose Pharma Inc. for providing EZN-2208.

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
