## [Reviewer comments · Open Biology]

Review History

RSOB-19-0262.R0 (Original submission)

Review form: Reviewer 1

Recommendation

Major revision is needed (please make suggestions in comments)

Do you have any ethical concerns with this paper?

No

Comments to the Author

The manuscript by Valsecchi et al “EZN-2208 treatment suppresses CLL by interfering with environmental protection and outweighs response to fludarabine” investigates the activity of the camptothecin-11 analog EZN-2208 as an antitumoral agent and its capacity to improve the response to the purine analog fludarabine in preclinical models of chronic lymphocytic leukemia (CLL). For this aim authors evaluated in vitro the apoptogenic effect of the compound when added to fludarabine, both at fixed doses, in CLL primary cells co-cultured with the bone-

marrow derived mesenchymal cell line HS-5. They further tested this combination in immunocompetent mice injected with splenic cells from Eu-TCL1 leukemic mice and in immunocompromised mice injected with the human CLL cell line MEC-1. They found that at a non-toxic dose, EZN-2208 cooperates with fludarabine in inducing CLL cell death, only in the case of CLL-HS-5 co-culture. In *in vivo* settings, authors found that a single cycle of treatment of TCL1-injected mice with EZN-2208 was enough to significantly decrease the number of CLL cells in the bone marrow and to lower the weight of the spleen, after fludarabine co-administration. However, after two cycles of EZN-2208 treatment, the activity of the combination was apparently mainly due to the sole compound, either in the TCL1 model or in the MEC-1 xenografts. In general, this is a well-structured and well-written manuscript which provides insights into the possible use of camptothecin derivatives in combination with classical purine analog therapy in CLL. Although they are not complemented by in deep functional analysis, the efficacy results shown here have been collected in validated *in vitro* and *in vivo* models, making most of author's conclusions based on consistent data. However, a number of issues should be addressed by the authors to improve the soundness of this work.

- Abstract: in the absence of data about HIF-1a inhibition in the presence manuscript, authors should avoid to mention this factor here.
- Figure 1: several methodological information is missing here. When authors claim that data represent the mean values of 11/8 experiments, they should clearly indicate the number of CLL primary samples analyzed and the number of replicates for each sample. Further, in figures 1A and 1C the duration of the treatment should be provided, and the method used to normalize the % of apoptotic cells to untreated cells clearly stated (the simple subtraction of basal apoptosis is not acceptable). Finally, it appears that Fig 1C and 1D have been interchanged in the figure legend section (according to data and text, Fig 1C should refer to CLL-HS5 co-cultures). Please correct.
- Figure 2: statistical analysis should be provided for blood sample analysis in the different mouse models and in the survival analysis (figure 2D). Most importantly, in the absence of robust dose-response, PK or PD analysis, authors are not allowed to compare the activity of EZN-2208 with fludarabine, and therefore their statement that EZN-2208 outweighs response to fludarabine is not supported by the present data. They should rather comment on the improvement of fludarabine response upon EZN-2208 co-administration and correct the corresponding text (title, abstract and result sections).
- In the absence of robust drug interaction analysis (Chou and Talalay algorithm), authors are not allowed to employ terms like "higher-than-additive" (page 4) or "fludarabine synergize *in vitro*" (page 5). Please correct.

Decision letter (RSOB-19-0262.R0)

05-Feb-2020

Dear Dr Bernardi,

We are writing to inform you that the Editor has reached a decision on your manuscript RSOB-19-0262 entitled "EZN-2208 treatment suppresses CLL by interfering with environmental protection and outweighs response to fludarabine", submitted to Open Biology.

As you will see from the reviewer's comments below, there are a number of criticisms that prevent us from accepting your manuscript at this stage. The reviewer suggests, however, that a revised version could be acceptable, if you are able to address their concerns. If you think that you can deal satisfactorily with the reviewer's suggestions, we would be pleased to consider a revised manuscript.

The revision will be re-reviewed, where possible, by the original referees. As such, please submit the revised version of your manuscript within four weeks. If you do not think you will be able to meet this date please let us know immediately.

When submitting your revised manuscript, please respond to the comments made by the referee and upload a file "Response to Referees" in "Section 6 - File Upload". You can use this to document any changes you make to the original manuscript. In order to expedite the processing of the revised manuscript, please be as specific as possible in your response to the referee(s).

Please see our detailed instructions for revision requirements
<https://royalsociety.org/journals/authors/author-guidelines/>

Sincerely,

The Open Biology Team
mailto: openbiology@royalsociety.org

Reviewer's Comments to Author:

Referee:

Comments to the Author(s)

The manuscript by Valsecchi et al "EZN-2208 treatment suppresses CLL by interfering with environmental protection and outweighs response to fludarabine" investigates the activity of the camptothecin-11 analog EZN-2208 as an antitumoral agent and its capacity to improve the response to the purine analog fludarabine in preclinical models of chronic lymphocytic leukemia (CLL). For this aim authors evaluated in vitro the apoptogenic effect of the compound when added to fludarabine, both at fixed doses, in CLL primary cells co-cultured with the bone-marrow derived mesenchymal cell line HS-5. They further tested this combination in immunocompetent mice injected with splenic cells from Eu-TCL1 leukemic mice and in immunocompromised mice injected with the human CLL cell line MEC-1. They found that at a non-toxic dose, EZN-2208 cooperates with fludarabine in inducing CLL cell death, only in the case of CLL-HS-5 co-culture. In in vivo settings, authors found that a single cycle of treatment of TCL1-injected mice with EZN-2208 was enough to significantly decrease the number of CLL cells in the bone marrow and to lower the weight of the spleen, after fludarabine co-administration. However, after two cycles of EZN-2208 treatment, the activity of the combination was apparently mainly due to the sole compound, either in the TCL1 model or in the MEC-1 xenografts. In general, this is a well-structured and well-written manuscript which provides insights into the possible use of camptothecin derivatives in combination with classical purine analog therapy in CLL. Although they are not complemented by in deep functional analysis, the efficacy results shown here have been collected in validated in vitro and in vivo models, making most of author's conclusions based on consistent data. However, a number of issues should be addressed by the authors to improve the soundness of this work.

- Abstract: in the absence of data about HIF-1a inhibition in the presence manuscript, authors should avoid to mention this factor here.
- Figure 1: several methodological information is missing here. When authors claim that data represent the mean values of 11/8 experiments, they should clearly indicate the number of CLL primary samples analyzed and the number of replicates for each sample. Further, in figures 1A and 1C the duration of the treatment should be provided, and the method used to normalize the % of apoptotic cells to untreated cells clearly stated (the simple subtraction of basal apoptosis is not acceptable). Finally, it appears that Fig 1C and 1D have been interchanged in the figure legend section (according to data and text, Fig 1C should refer to CLL-HS5 co-cultures). Please correct.
- Figure 2: statistical analysis should be provided for blood sample analysis in the different mouse models and in the survival analysis (figure 2D). Most importantly, in the absence of robust dose-response, PK or PD analysis, authors are not allowed to compare the activity of EZN-2208 with fludarabine, and therefore their statement that EZN-2208 outweighs response to fludarabine is not supported by the present data. They should rather comment on the improvement of fludarabine response upon EZN-2208 co-administration and correct the corresponding text (title, abstract and result sections).
- In the absence of robust drug interaction analysis (Chou and Talalay algorithm), authors are not allowed to employ terms like "higher-than-additive" (page 4) or "fludarabine synergize in vitro" (page 5). Please correct.

Author's Response to Decision Letter for (RSOB-19-0262.R0)

See Appendix A.

RSOB-19-0262.R1 (Revision)

Review form: Reviewer 1

Recommendation

Accept as is

Do you have any ethical concerns with this paper?

No

Comments to the Author

Authors have adequately addressed the different issues raised by my previous review.

Review form: Reviewer 2

Recommendation

Major revision is needed (please make suggestions in comments)

Do you have any ethical concerns with this paper?

No

Comments to the Author

Chronic lymphocytic leukemia is a lymphoproliferative disease that has not cure. One of the most important concerns in the treatment of this disease is the high percentage of patients developing resistance. Because of that, the search of new compounds to be used alone or in combination with the current therapies is necessary. In this paper, authors show a new compound, EZN-2208, as a good candidate for the treatment of refractory CLL. EZN-2208 shows cytotoxicity against CLL cells from patients and increases response to fludarabine. The topic of this paper is significant, but important major issues remain to be addressed:

Major concerns:

1) Figure 1:

- In Figure 1A, C and D, the real percentage of annexin V+ cells must be shown, as in Figure B, since it is important to show the percentage of spontaneous apoptosis and the real effect of EZN-2208 in cell death.

- In Figure 1B, is the increase in annexin V+ cells in the condition with EZN-2208 significant with respect to control condition? Because in Figure A the increase is significant and number of samples used are very similar.

- In Figure C and D authors must show the p value comparing untreated vs combination, it is supposed to be significant.

- To better understand the contribution of EZN-2208 to the fludarabine-induced cytotoxicity, a synergy study should be performed with an appropriate statistical analysis.

2) Figure 2:

- In most of graphics in Figure 2 authors do not show the p value comparing Untreated vs Combination. Is it because these p values are not significant? Authors must show all p values in order to understand which of them are significant or not because it seems obvious that some missing p values are significant considering the S.E.M., for example, in 2A left panel or in 2C middle panel.

- Why authors have used higher doses of EZN-2208 in in vivo experiments with the E μ -TCL1-derived CLL model than in the MEC-1 model? As they say in the text, they reduce the doses administrated to study better the cooperative effects with fludarabine, the same strategy should have followed in the other model. How do they explain the different strategies for the in vivo experiments?

- Show the survival curves in 2A and 2B.

3) The total number of CLL patients used in this paper should be reported, and a table with the clinical characteristics is recommendable.

Decision letter (RSOB-19-0262.R1)

09-Mar-2020

Dear Dr Bernardi,

We are writing to inform you that the Editor has reached a decision on your manuscript RSOB-19-0262.R1 entitled "EZN-2208 treatment suppresses CLL by interfering with environmental protection and increases response to fludarabine", submitted to Open Biology.

As you will see from the reviewers' comments below, there are a number of criticisms that prevent us from accepting your manuscript at this stage. The reviewers suggest, however, that a revised version could be acceptable, if you are able to address their concerns. If you think that you can deal satisfactorily with the reviewer's suggestions, we would be pleased to consider a revised manuscript.

The revision will be re-reviewed, where possible, by the original referees. As such, please submit the revised version of your manuscript within six weeks. If you do not think you will be able to meet this date please let us know immediately.

When submitting your revised manuscript, please respond to the comments made by the referee(s) and upload a file "Response to Referees" in "Section 6 - File Upload". You can use this to document any changes you make to the original manuscript. In order to expedite the processing of the revised manuscript, please be as specific as possible in your response to the referee(s).

Please see our detailed instructions for revision requirements. It is essential these instructions are followed carefully to minimize any delay to publication:

<https://royalsociety.org/journals/authors/author-guidelines/>

Sincerely,
The Open Biology Team
mailto: openbiology@royalsociety.org

Reviewer(s)' Comments to Author(s):

Referee: 1

Comments to the Author(s)

Authors have adequately addressed the different issues raised by my previous review.

Referee: 2

Comments to the Author(s)

Chronic lymphocytic leukemia is a lymphoproliferative disease that has not cure. One of the most important concerns in the treatment of this disease is the high percentage of patients developing resistance. Because of that, the search of new compounds to be used alone or in combination with the current therapies is necessary. In this paper, authors show a new compound, EZN-2208, as a good candidate for the treatment of refractory CLL. EZN-2208 shows cytotoxicity against CLL cells from patients and increases response to fludarabine. The topic of this paper is significant, but important major issues remain to be addressed:

Major concerns:

1) Figure 1:

- In Figure 1A, C and D, the real percentage of annexin V+ cells must be shown, as in Figure B, since it is important to show the percentage of spontaneous apoptosis and the real effect of EZN-2208 in cell death.

- In Figure 1B, is the increase in annexin V+ cells in the condition with EZN-2208 significant with respect to control condition? Because in Figure A the increase is significant and number of samples used are very similar.

- In Figure C and D authors must show the p value comparing untreated vs combination, it is supposed to be significant.
 - To better understand the contribution of EZN-2208 to the fludarabine-induced cytotoxicity, a synergy study should be performed with an appropriate statistical analysis.
- 2) Figure 2:
- In most of graphics in Figure 2 authors do not show the p value comparing Untreated vs Combination. Is it because these p values are not significant? Authors must show all p values in order to understand which of them are significant or not because it seems obvious that some missing p values are significant considering the S.E.M., for example, in 2A left panel or in 2C middle panel.
 - Why authors have used higher doses of EZN-2208 in in vivo experiments with the E μ -TCL1-derived CLL model than in the MEC-1 model? As they say in the text, they reduce the doses administrated to study better the cooperative effects with fludarabine, the same strategy should have followed in the other model. How do they explain the different strategies for the in vivo experiments?
 - Show the survival curves in 2A and 2B.
- 3) The total number of CLL patients used in this paper should be reported, and a table with the clinical characteristics is recommendable.

Author's Response to Decision Letter for (RSOB-19-0262.R1)

See Appendix B.

RSOB-19-0262.R2 (Revision)

Review form: Reviewer 2

Recommendation

Accept as is

Do you have any ethical concerns with this paper?

No

Comments to the Author

The authors have done important changes in the manuscript, improvement it and facilitating its comprehension. They have performed almost every change required; for that, I think that the manuscript is now appropriated to be published.

Decision letter (RSOB-19-0262.R2)

14-Apr-2020

Dear Dr Bernardi

We are pleased to inform you that your manuscript entitled "EZN-2208 treatment suppresses CLL by interfering with environmental protection and increases response to fludarabine" has been accepted by the Editor for publication in Open Biology.

If applicable, please find the referee comments below. No further changes are recommended.

Sincerely,
The Open Biology Team
mailto: openbiology@royalsociety.org

Reviewer(s)' Comments to Author:

Referee: 2

Comments to the Author(s)

The authors have done important changes in the manuscript, improvement it and facilitating its comprehension. They have performed almost every change required; for that, I think that the manuscript is now appropriated to be published.

Appendix A

Rebuttal letter

We thank this reviewer for his kind comments and suggestions. We agree with all his/her criticisms and suggestions and have modified our manuscript accordingly. Please, find below a detailed description of changes that have been introduced.

1. As suggested by the reviewer, we have avoided any sentence directly mentioning HIF-1 α inhibition by EZN-2208, because we are not showing data to prove this point in the current manuscript. Specifically, in the abstract we have substituted “a pharmacological compound that inhibits HIF-1 α ” with “a pharmacological compound previously shown to inhibit HIF-1 α ”, and “thus suggesting that HIF-1 α inhibitors” with “thus suggesting that this or similar compounds”.
2. Figure 1. We agree with the reviewer that our previous description of the experiments in the figure legend was not clear. By 8-11 independent experiments we meant 8-11 different primary samples, processes independently in different days. We have now corrected the figure legend accordingly. Due to the high number of cells needed for this experiment, samples were not seeded and processed in duplicate or triplicate wells. Regarding duration of treatment for panels A and C, like the other two panels cells were treated for 48 hours. This information has now been added. Regarding the correction method used to normalize cell death in treated cells over spontaneous cell death, the initial analysis had been performed by subtracting basal cell death. However, we agree with this reviewer that this is not the correct way to analyze the data, and we thank the reviewer for pointing this out. We have now corrected it in the new Figure 1 by calculating the ratio between annexin V positive cells in treated versus untreated cells (Figures 1A, C and D). The numbers have changed slightly, but the outcome of the experiments remains the same. Also, we have added a more detailed description of the normalization procedure in the Materials and Methods. Finally, the reviewer is right that panel C and D had been interchanged. They are now in the correct order.
3. Figure 2. Statistical analysis for blood samples in the different mouse models and in the survival plot has been performed again, as requested by this reviewer. Regarding blood analysis, we found only one significant difference that we had missed in our previous analysis, namely in the peripheral blood of EZN-2208 treated mice in Figure 2C. Everything else, where not indicated, is not statistically different. In the survival analysis in Figure 2D, we had omitted statistical analysis because we did not find any

difference between EZN-2208 and combination treatment. However, as requested by the reviewer, we have now added in the Figure legend the statistical analyses. The second point of this reviewer, namely that because we have not performed careful and comparative drug-response analyses in vivo we cannot compare the efficacy of EZN-2208 with that of fludarabine, is also well-taken. In agreement with the reviewer's suggestions we have thus avoided use of the term "outweigh" or similar terms when referring to EZN-2208 activity versus fludarabine. As suggested, we have corrected the text (title, abstract and main text) with a more accurate phrasing.

4. We agree with the reviewer that because we have not performed drug interaction tests we cannot describe our data using the terms "synergism" or "higher-than-additive" responses. We have thus removed these terms and stated instead that EZN-2208 increases cell mortality induced by fludarabine under certain experimental conditions, and not others.

Appendix B

Rebuttal letter

We thank both reviewers for their previous and current kind comments and suggestions.

Reviewer 1 is satisfied with our responses to his/her criticisms, while reviewer 2 recognizes the relevance of our work but raises a number of concerns that we have addressed as follows:

Figure 1

1. The reviewer asks that we show the real percentage of annexin V+ cells in all figure panels, to assess the effect of EZN-2208 on total cell death instead of normalizing over untreated samples, which undergo spontaneous cell death. We agree with this reviewer that this is a more accurate way of representing cell death induced by EZN-2208, and have modified Figure 1 accordingly.
2. The reviewer asks whether in Figure 1B the increase in annexin V+ cells in cells treated with EZN-2208 is significant with respect to control cells, as in our previous analysis it was significant in Figure 1A. However, when expressing cell death as real percentages of annexin V+ cells, rather than normalized values, treatment with 50 nM EZN-2208 does not induce significant cell death in either condition. We have corrected the text accordingly.
3. The reviewer asks that in panel C and D we show the p values obtained when comparing untreated cells vs cells treated with drug combination, expecting the differences to be significant. We have now shown all significant differences, and confirm that dual drug treatment induces significant cell death when compared to untreated cells.
4. This reviewer suggests that we perform a synergy study to better understand the contribution of EZN-2208 to fludarabine-induced cytotoxicity. We agree that such a study would allow us to better delineate drug interactions and formally demonstrate drug synergism. However, our collaboration with Enzon Pharmaceuticals, the company that developed EZN-2208 and provided it to us on a collaborative basis, was interrupted when Belrose Pharma acquired Enzon, and we have been unable to obtain new batches of EZN-2208 since. Therefore unfortunately we are unable to perform the experiment suggested by this reviewer.

Figure 2:

1. The reviewer asks that we show all significant p values, as in most panels we had not included p values comparing untreated cells versus cells treated with the drug combination. As observed by the reviewer, in most cases differences between untreated mice and mice treated with the drug combination are indeed significant. We have now added all significant p values. Where not indicated, differences are not significant.
2. The reviewer asks why we have used different doses of EZN-2208 for the in vivo experiments with MEC1 cells (2 mg/kg) and Tc11 mice-derived cells (5 mg/kg). His/her question is very well taken. The reason is that we had indeed used the dose of 5 mg/kg EZN-2208 in combination with fludarabine in a first experiment with MEC1 cells, without seeing any additive effect, as reported in Figure 1 of this rebuttal letter.

However, because the dose of 5 mg/kg EZN-2208 slows MEC1-driven disease quite effectively (Figure 1), we hypothesized that lowering EZN-2208 dosage would allow more space to uncover any cooperative effect of the co-treatment with fludarabine, albeit this experiment also failed to show significant differences (Figure 2 of the manuscript). We have now provided this explanation in the text.

Conversely, for the Tc11 transplantation experiment, we used a slow-progressing leukemia that gave us enough space to test combination treatments. For this reason we chose to use the standard dose of 5 mg/kg of EZN-2208 that we had previously characterized (Valsecchi et al., 2016). However, since we did not observe any added efficacy in all the survival experiments that we have performed, we are ready to remove the survival curve from Figure 2, if this reviewers deems it appropriate.

- The reviewer asks that we show the survival curves for Figure 2A and B. We do not have survival curves for these two panels, as all the mice were sacrificed at the end of one or two cycles of treatment (day 43 and 63 respectively), as indicated in the text. However, a third cohort of mice was analyzed for survival after two cycles of treatment, which is indicated in the text as data not shown. As requested by this reviewer, we provide the survival analysis in this rebuttal letter (Figure 2), which shows no significant

difference when mice were treated with the drug combination as compared to the single drugs. We are inclined not to present these data in the paper because, due to the natural slow course of this disease, the mice survived for a long time after the second cycle of treatment (from day 63 to day 100 and over) and the effect of acute drug treatment may have been lost or modified over time, giving rise to results that are not linear. Also, as discussed earlier, given that all survival curves gave negative results, we may remove them all and just mention the results in the text. Nonetheless, if this reviewer believes that showing the data may be useful, we are ready to include them in the manuscript.

Finally, as requested by the reviewer, we have included a table reporting the total number of patients samples used in this manuscript and the available clinical characteristics (Table 1).